# *Schistosoma mansoni* soluble egg antigen and its key proteins differentially affect dextran sodium sulphate-induced inflammatory bowel disease

**Hsiang-Wei Fan[1], Ho Yin Pekkle Lam[1,2,3]/+**

[1]Tzu Chi University, School of Medicine, Master Program in Biomedical Sciences, Hualien, Taiwan
[2]Tzu Chi University, School of Medicine, Department of Biochemistry, Hualien, Taiwan
[3]Tzu Chi University, Institute of Medical Science, Hualien, Taiwan

**BACKGROUND** Inflammatory bowel disease (IBD) is an increasingly prevalent disease, affecting over seven million people worldwide and imposes a heavy burden on public health. The rising prevalence of IBD may be attributed to the hygiene hypothesis, which suggests that reduced exposure to parasites and microbes may weaken the immune system, thereby increasing susceptibility to developing IBD. Studies suggest helminths and their secretory products can modulate the host immunity and attenuate IBD. Our previous research also demonstrated that intestinal schistosomiasis can mitigate chronic IBD symptoms by restoring intestinal immune balance and dysbiosis.

**OBJECTIVES** While the primary pathology of schistosomiasis results from egg entrapment, we hypothesised that soluble egg antigen (SEA), known for its strong immunomodulatory effect, may contribute to the improvement of IBD. Given that SEA comprises multiple different proteins, identifying the role of individual components may clarify the therapeutic potential of SEA in IBD.

**METHODS** BALB/c mice were induced with dextran sodium sulphate (DSS) to develop IBD. Throughout the experiment, mice were intraperitoneally injected with 250 µg/mL crude SEA extract or recombinant egg antigen proteins, including SM14, GST28, and SMP40, three times a week. Colonic histopathology was assessed by H&E staining, and the immune response was evaluated through periodic acid-Schiff (PAS) staining, immunohistochemistry, enzyme-linked immunosorbent assay (ELISA), western blot, and quantitative polymerase chain reaction (qPCR).

**FINDINGS** Both SEA and Smp40 alleviated DSS-induced IBD, whereas SM14 exacerbated the disease and led to colonic dysplasia. In contrast, GST28 showed no significant effect on IBD. Further investigation revealed that all tested proteins modulated the immune response in mice, though each did so in different ways. These differences in immune modulation may underlie the varying disease outcomes observed.

**MAIN CONCLUSIONS** While SEA has shown therapeutic promise in IBD, it is also important to investigate the safety and mechanisms of individual antigens before considering their clinical application in the future.

Key words: inflammatory bowel disease - soluble egg antigen - *Schistosoma* - colonic dysplasia

Inflammatory bowel disease (IBD) is an increasingly prevalent chronic inflammatory condition that currently affects over seven million people worldwide.[1] The disease places a substantial burden on public health, leading to high medical costs, reduced productivity, and diminished quality of life.[1] The etiology of IBD is not clear, but imbalanced immunity, especially the polarisation toward the Th1 response and heightened inflammation, has been suggested as a primary reason leading to chronic intestinal inflammation and IBD.[2] In the 21st century, the incidence of IBD has accelerated dramatically in both developing and developed countries. The steep increase in IBD incidence was explained by the hygiene hypothesis, which suggests that people are less likely to be exposed to microorganisms with greater urbanisation and improved hygiene practices. As a result, children may lead to an inadequate maturation of the immune system and loss of its ability to modulate the autoimmune response.[3,4]

It was suggested that several helminths, such as *Schistosoma mansoni*,[5,6] *Trichuris suis*,[7,8] hookworms,[9] and *Strongyloides venezuelensis*,[10] can modulate the immunity and attenuates IBD. Among the diseases caused by these parasites, schistosomiasis ranks as the second most important parasitic disease after malaria, affecting more than 200 million people around the world.[11] Despite its prevalence, our previous research suggests that intestinal schistosomiasis can alleviate IBD by reversing intestinal immune imbalance and gut dysbiosis.[6] Because the primary pathology of schistosomiasis results from egg entrapment, the observed improvement in colitis may be attributed to proteins secreted by the

Financial support: This research was funded by the National Science and Technology Council of Taiwan (grant number NSTC 113-2320-B-320-003-MY3).
+ Corresponding author: pekklelavabo@gms.tcu.edu.tw | ⬮ https://orcid.org/0000-0001-8378-7811

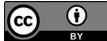

eggs, which are known to exert strong immunomodulatory effects.[12] In natural infection, the trapped eggs continuously secrete soluble egg antigen (SEA), which stimulates the host's T helper 2 (Th2) immune response, characterised by an increase of IL-4, IL-5, and IL-13.[13] This Th2 response exerts anti-inflammatory effects that protect the host from excessive inflammation; however, it also contributes to the pathology of schistosomiasis by promoting egg-induced granulomas and fibrosis.[13,14] When used in isolation, SEA functions as a potent immune modulator and has been explored for its therapeutic potential in autoimmune and inflammatory diseases.[12] Although the direct effect of SEA on colitis has yet to be studied, exosomes derived from dendritic cells treated with SEA have been shown to prevent acute colitis by suppressing inflammatory cytokines.[15]

While the term "SEA" refers to "soluble egg antigen", it does not represent a single, defined antigen. Instead, SEA is a crude extract derived from schistosome eggs, comprising a heterogeneous mixture of biomolecules, including proteins, glycoproteins, polysaccharides, and glycolipids.[16] These components originate from various egg-associated sources, such as the eggshell, the miracidium within the egg, and secretory products released by the egg.[17] Currently, several SEA-associated antigens have been discovered, including *S. mansoni* 14-kDa fatty acid-binding protein (SM14), 28-kDa glutathione S-transferase (GST28), and 40-kDa heat shock protein (major egg protein; Smp40). These proteins have previously demonstrated strong immunogenicity,[18,19,20,21] with SM14 and GST28 being developed as vaccine candidates and advancing into human clinical trials.[22,23] GST28 has also been shown to ameliorate experimental colitis by modulating the host immune response.[24,25] In a phase IIA clinical study, subcutaneous injections of GST28 led to a decrease in the disease activity index in patients with Crohn's disease, with only minimal GST28-related adverse effects observed.[26] However, limited by the sample size in that study (only 10 patients enrolled),[26] further evidence is needed to clarify the therapeutic effect of GST28 in IBD. Yet, these findings suggest a beneficial role of egg antigens and highlight their potential as a therapeutic agent for IBD.

In this study, we aimed to investigate the effect of crude SEA, along with recombinant SM14, GST28, and Smp40, in a dextran sodium sulphate (DSS)-induced IBD model. Additionally, we sought to clarify the immunomodulatory roles of these proteins in the context of IBD.

## MATERIALS AND METHODS

*Animals and parasites* - Animal experiments were approved by the Institutional Animal Care and Use Committees (IACUC) of Tzu Chi University (No. 113043). Male BALB/c mice were purchased from the National Laboratory Animal Centre (NLAC), NARLabs, Taiwan. All mice were housed under a 23ºC ± 1ºC and a 12-h light/dark cycle condition with 40-60% humidity. Food and water were available *ad libitum*.

Puerto Rico strain of *S. mansoni* was provided by the Biomedical Research Institute, MD, USA. The freshwater snail *Biomphalaria glabrata* was used as an intermediate host and male BALB/c mice were used as the final host. The *S. mansoni* life cycle was maintained as described previously.[6]

*Preparation of SEA* - Mice infected with *S. mansoni* were sacrificed eight weeks post-infection. Livers were collected and homogenised in ice-cold phosphate-buffered saline (PBS). Eggs were isolated from the liver homogenates by sequential filtration through a series of sieves with decreasing pore size: 420, 177, 105, and 25 μm. Eggs retained on the 25 μm sieve were collected in ice-cold PBS and centrifuged at 370× g for 2 min. The pellet was resuspended in ice-cold PBS, and SEA was prepared by homogenising the eggs with a glass homogeniser. All procedures were conducted under sterile conditions.

*Preparation of recombinant SM14, GST28, and Smp40* - Plasmids of the SM14 and GST28 were constructed as previously described.[19,27] The plasmid of the Smp40 was newly designed and constructed in this experiment. Briefly, a polymerase chain reaction (PCR) was performed on the cDNA of *S. mansoni* worm with the primer: Forward 5' GAGACATATGTCTGGTGG-GAAACAACATAAC 3' and Reverse 5' GATACTC-GAGGTGAGTAATTGCATGTTGCTTC 3'. The PCR product was run on a 1% agarose gel and purified by EasyPure PCR/Gel extraction Kit (Bioman Scientific, Taipei, Taiwan). The resulting product was digested with NdeI and XhoI and ligated into plasmid pET-28a [Supplementary data (Figure A)].

All plasmids were transformed into the *Escherichia coli* BL21 (DE3) for protein expression. Transformed BL21 (DE3) was plated onto LB agar containing 50 μg/mL kanamycin and incubated overnight. Single colonies were inoculated into LB broth and cultured at 37ºC with shaking until an $OD_{600}$ of 0.8 was reached. Protein expression was induced by adding 1 mM isopropyl-beta-D-thiogalactopyranoside (IPTG), followed by incubation at 37ºC with shaking for 4 h. Cultures were harvested by centrifugation at $10,000 \times g$ for 20 min at 4ºC. The supernatant was discarded, and the remaining pellets were resuspended in lysis buffer (50 mM NaH$_2$PO$_4$, 300 mM NaCl, 10 mM imidazole, pH 8). The cells were lysed using a MiniBead-beater and the lysate was collected and centrifuged at $20,000 \times g$ for 15 min at 4ºC. The supernatant containing soluble proteins was collected for purification by metal affinity chromatography (Cat#: 7880011; Bio-Rad, CA, USA). The eluted proteins were analysed on a 13.5% sodium dodecyl sulphate-polyacrylamide gel (SDS-PAGE) and stained with 0.1% Coomassie blue. An SDS-PAGE gel showing the bacterial lysates for Smp40 with and without IPTG induction, as well as the eluted Smp40 protein, is presented in Supplementary data (Figure B-D).

*Animal treatment* - Mice were six weeks of age at the beginning of the experiments. Mice were divided randomly into five groups. The sample size was calculated using the resource equation, resulting in the allocation of three to five mice per group. The experiment was independently repeated twice. All the mice were given 2% DSS (Cat#: J63606.22, Thermo Fisher Scientific,

MA, USA) in their drinking water at week 1 and week 3, whereas normal drinking water was provided at week 2 and week 3. Different egg antigen proteins (at a concentration of 250 μg/mL) were peritoneally injected into the mice three times a week on alternate days during the experimental duration. All the mice were sacrificed after four weeks (Fig. 1). At the time of sacrifice, blood was collected through cardiac puncture, and organs were collected for subsequent experiments.

*Disease activity index (DAI)* - The DAI of the mice was assessed weekly based on three criteria: percentage of weight loss, presence of rectal bleeding, and stool consistency. Each parameter was scored on a scale from 0 to 4 [Supplementary data (Table I)], and the DAI score was calculated as the sum of the three individual scores.

*Tissue processing, staining, and histopathology* - Tissues were fixed with 10% formalin, embedded in paraffin, and sectioned into thin slices for haematoxylin & eosin (H&E) and periodic acid-Schiff (PAS) staining as previously described.[6] Colonic sections stained with H&E were scored for epithelial damage, lamina propria inflammation, muscularis propria thickening, and fibrosis. Each criterion was assigned a score of 0, not observed; 1, mild; 2, moderate; 3, intensive. Measurements were also done on villous height, crypt depth, and villi-to-crypt ratio. Colonic sections stained with PAS were counted for the number of positive cells. At least ten random fields were examined and scored in each section.

Colonic sections were also stained for MUC-2 by immunohistochemistry staining. Briefly, antigens were retrieved from the sections by incubating them in boiling EDTA buffer for 20 min. Subsequently, the sections were treated with 3% $H_2O_2$ for 10 min and incubated overnight at 4ºC with MUC-2 (1:300; Cat#: A14659; ABclonal). The sections were then incubated with HRP-conjugated anti-rabbit secondary antibody (1:1000; Cat#: C04003; Croyez Biosciences) for 30 min and 3, 30-diaminobenzidine (DAB; Cat#: 34000;

Thermo Fisher Scientific) for 12 min. Sections were counterstained with haematoxylin and rehydrated with increasing ethanol concentration.

*RNA extraction, cDNA synthesis, and quantitative PCR (qPCR)* - Total RNA was extracted by homogenising tissues in TRIzol reagent (Invitrogen; Thermo Fisher Scientific) and purified using the standard chloroform extraction method. Five micrograms of total RNA were used to generate cDNA using a GScript First-Strand Synthesis Kit (GeneDireX, Taiwan). The qPCR reaction was performed by 2× qPCRBIO SyGreen Blue Mix Lo-ROX (PCR Biosystems, London, UK) using the Roche LightCycler 480 system. Amplification and detection were performed as follows: 55 cycles of denaturation at 95ºC for 10 s, 58 or 60ºC for 15 s, and extension at 72ºC for 25 s. The oligonucleotide primers used are shown in Supplementary data (Table II). Relative gene expression was calculated using the $2^{-\Delta\Delta CT}$ method with β-actin as the housekeeping gene.

*Measurement of cytokine levels in serum and colonic tissues* - Serum was separated by centrifuging whole blood at 1,500 × g for 15 min. The colon was homogenised in PBS and centrifuged at 12,000 × g for 15 min at 4ºC. Levels of IL-1β (Cat#: 432604; BioLegend, San Diego, CA, USA), IL-2 (Cat#: 431001; BioLegend), IFN-γ (Cat#: 430801; BioLegend), IL-4 (Cat#: 431101; BioLegend), IL-5 (Cat#: 431204; BioLegend), IL-17A (Cat#: 88-7371; Thermo Fisher Scientific), IL-22 (Cat#: 88-7422; Thermo Fisher Scientific), and IL-10 (Cat#: 431411; BioLegend) in the sera or colonic homogenate were measured using a standard sandwich enzyme-linked immunosorbent assay (ELISA) kit. Protein concentrations of colonic homogenates were determined by the Bradford method using a Bio-Rad Protein Assay Dye (Bio-Rad Laboratories, Hercules, CA, USA).

*Western blot* - Total proteins were extracted, separated by 10% SDS-PAGE, and transferred onto PVDF membranes (EMD Millipore, Burlington, MA, USA). After blocking with 5% non-fat milk, the membranes

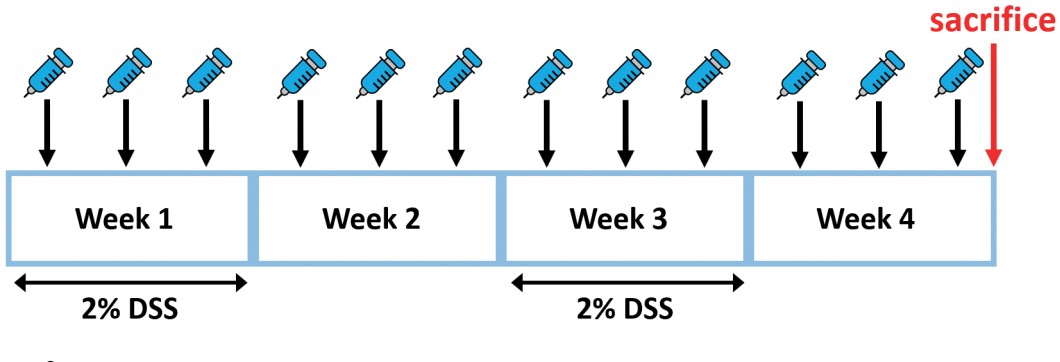

Fig. 1: experimental scheme. Mice were given 2% dextran sulphate sodium (DSS) in their drinking water at weeks 1 and 3, and normal drinking water was provided at weeks 2 and 4. Different egg antigens (at a concentration of 250 μg/mL) were peritoneally injected into the mice three times a week on alternate days during the experimental duration. All the mice were sacrificed after four weeks.

were incubated with MUC-2 (1:1000; Cat#: A14659; Abclonal) or α-tubulin (1:5000; Cat#: AC012; Abclonal) at 4ºC overnight. Membranes were then incubated with HRP-conjugated anti-rabbit (1:5000; Cat#: C04003; Croyez Biosciences) or HRP-conjugated anti-mouse (1:5000; Cat#: C04001; Croyez Biosciences) secondary antibody for 1 h. Membranes were developed using ECL detection reagent (EMD Millipore). Relative protein levels were quantified using Image J (Version 1.46, National Institute of Health, Bethesda, MD, USA), and protein densitometry was expressed relative to that of α-tubulin.

*Statistical analysis* - Two independent experimental replicates were performed. Data from the same groups across both repeats were pooled, and a statistical comparison was performed on the combined dataset without applying interim statistical adjustments. Data are presented as mean ± standard deviation (SD) unless otherwise specified. Statistical significance between groups was evaluated using one-way analysis of variance (ANOVA) followed by Tukey's honest significant difference test. Differences were considered statistically significant at $p < 0.05$ (*$p < 0.05$; **$p < 0.01$; ***$p < 0.001$; ****$p < 0.0001$). All analyses were performed by GraphPad Prism software version 9.4.1.

## RESULTS

*Soluble egg antigen and Smp40 improve DSS-induced colitis, whereas SM14 worsens DSS-induced colitis* - Mice were treated with two repeated cycles of 2% DSS to develop colitis. At the same time, mice were intraperitoneally injected with either crude SEA or re-combinant egg antigens, SM14, GST28, and Smp40, at a concentration of 250 µg/mL three times a week (Fig. 1). Injection of different egg antigens had minimal but different effects on the body weight of the mice. SEA- and Smp40-treated mice have improved body weight compared to the vehicle-treated mice, but only Smp40-treated mice reached statistical significance (Fig. 2A). Regarding the severity of colitis, mice treated with SM14 showed a slightly higher, although not statistically significant, DAI score (Fig. 2B) and shortened colon length (Fig. 2C-D). Histopathological analysis of the colon suggested inflammatory infiltration in all the groups, though it was notably reduced in the SEA-, GST-, and Smp40-treated mice. In contrast, SM14-treated mice showed increased inflammatory infiltrations and were the only group in which colonic dysplasia was observed (Fig. 3A). Histological analysis also revealed a worsened histological score in the SM14-treated group (Fig. 3B). Further analysis suggested that Smp40-treated mice exhibited increased villus length, while SM14-treated mice had a shorter villus length (Fig. 3C). While these antigens did not affect the depth of the crypt (Fig. 3D), Smp40-treated mice had a significantly higher villus-to-crypt ratio (Fig. 3E), suggesting improved intestinal architecture and functions. SM14-treated mice, on the other hand, exhibited a reduced villus-to-crypt ratio compared to other groups (Fig. 3E), indicating colonic damage and compromised intestinal function.

To further assess inflammation, colonic IL-1β levels were measured. SEA- and Smp40-treated groups exhibited decreased IL-1β levels, while the SM14-treat-

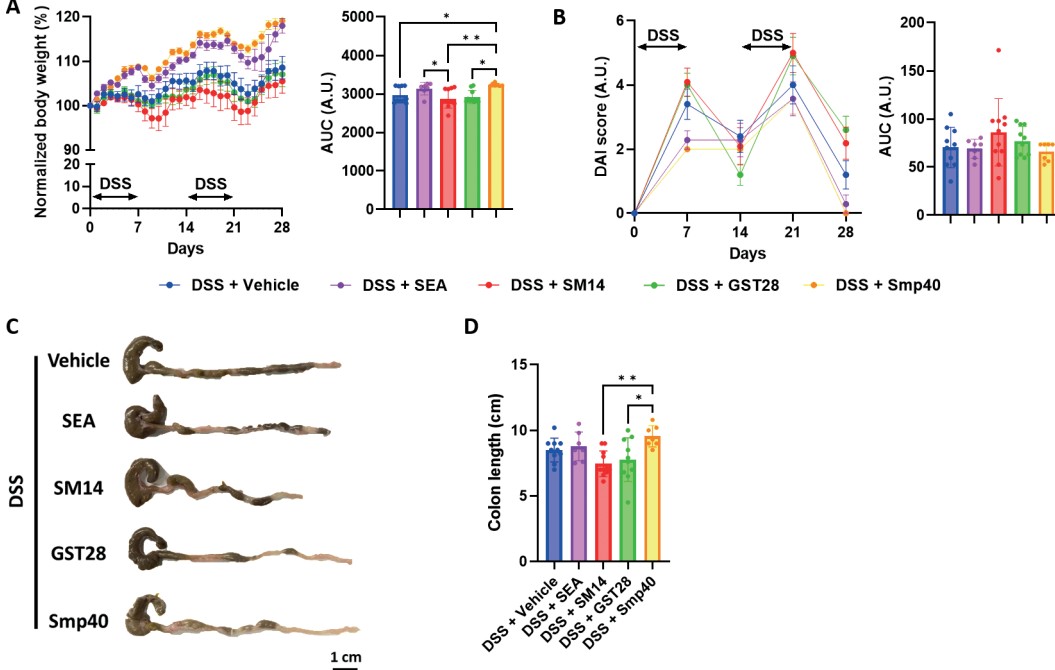

Fig. 2: schistosome egg antigen differently alters dextran sulphate sodium (DSS)-induced inflammatory bowel disease in mice. (A) Body weight change of the mice and the corresponding area under curve (AUC) chart. (B) Disease activity index and the corresponding AUC chart. (C) Representative colon images of the mice. (D) Colon length. $n$ = 7-11 mice. Data are presented as mean ± standard deviation (SD). *$p < 0.05$ and **$p < 0.01$. Significance determined by one-way analysis of variance (ANOVA).

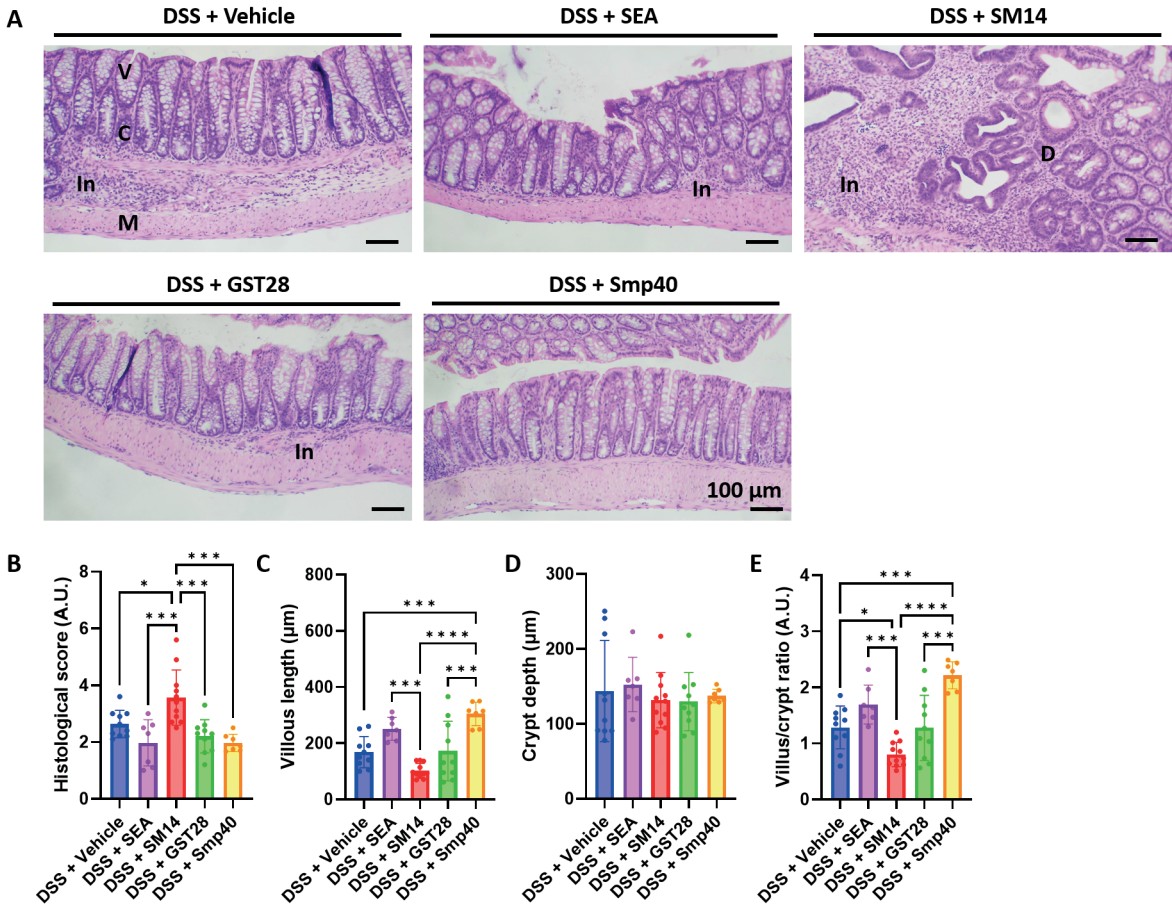

Fig. 3: schistosome egg antigen differently alters colon histopathology in dextran sulphate sodium (DSS)-induced colitis in mice. (A) Representative histological images of the colon tissue. V: villus; C: crypts; M: muscularis mucosae; In: inflammation; D: dysplasia. (B) Histological score of the colon. (C) Measurement of villi length, (D) crypt depth, and (E) villi-to-crypt ratio. *n* = 7-11 mice. Data are presented as mean ± standard deviation (SD). *p < 0.05; ***p < 0.001; and ****p < 0.0001. Significance determined by one-way analysis of variance (ANOVA).

ed group showed increased levels (Fig. 4A). Similarly, the SM14-treated group showed a higher serum IL-1β level compared to the other groups (Fig. 4B). Collectively, these results suggest that different schistosome egg antigens may exert differential effects on DSS-induced IBD. While SEA and Smp40 appear to alleviate colitis, SM14 worsens colitis.

*SEA improves intestinal barrier by increasing goblet cells and upregulating MUC-2 expression* - To further investigate the function of the intestinal barrier, PAS staining was used to identify mucin within goblet cells, which appeared dark pink in contrast to the pale pink background. The result suggested that SEA-treated mice had significantly higher colonic goblet cells in the villus compared to the vehicle and other treatment groups (Fig. 5A-B). Yet, goblet cell numbers were not significantly different in the crypt (Fig. 5C). Western blot analysis revealed increased MUC-2 expression in the SEA-treated group (Fig. 5D-E). However, immunohistochemistry staining for MUC-2 in colonic tissue revealed only minimal and non-significant changes (Fig. 5F-G). While SEA treatment showed a substantial effect on the intestinal barrier, changes in other treatment groups were not as evident as those in SEA-treated groups.

*Schistosome egg antigen differently modulates immune responses of colitic mice* - Given the critical role of immune imbalance in the pathogenesis of IBD[2] and the immunoregulatory effects of schistosome eggs and their antigens,[12] we next aimed to investigate the immune profile in the mice. Measuring the cytokines in the colon revealed significant suppression of IL-2, IFN-γ, IL-4, IL-5, and IL-10 levels in the SEA- and Smp40-treated groups compared to SM14- and GST28-treated groups. Although a noticeable decrease in the cytokine levels was also seen in the SEA- and Smp40-treated group compared to the vehicle-treated group, the results were statistical insignificant (Fig. 6A-D, G). In contrast, SM14-treated group showed a significant increase in IL-2, IFN-γ, IL-4, and IL-10, but only when compared to the SEA- and Smp40-treated groups (Fig. 6A-C, G). Additionally, GST28-treated group exhibited a significant increase in the colonic IL-2, IL-4, and IL-5 levels compared to the vehicle-treated group (Fig. 6A, C-D). It is worth noting that IL-17A and IL-22 levels were not altered by any of the egg antigens (Fig. 6E-F).

The immune profile in the spleen and serum was also analysed to assess systemic immunity. As noted, the Smp40-treated group showed a significant upregulation

of splenic expression of IFN-γ, IL-5, IL-22, and IL-10 (Fig. 7B, D, F-G). IL-2, although not showing statistical significance (p = 0.0597), also revealed a similar increase in the Smp40-treated group (Fig. 7A). The SM14-treated group also exhibited increased IL-22 expression compared to the vehicle-treated group (Fig. 7F), while the SEA-treated group significantly upregulated splenic IL-10 expression (Fig. 7G). On the other hand, SEA-treated group significantly suppressed serum IFN-γ and IL-10

levels while Smp40-treated group suppressed serum IFN-γ levels (Fig. 7I, N). In addition, the GST28-treated group exhibited an increase in serum IL-17A levels (Fig. 5L). These results suggest that distinct *Schistosoma* egg antigens can differentially modulate both local intestinal immune response and systemic immunity.

## DISCUSSION

Soluble egg antigen of *Schistosoma* has demonstrated good immunogenic properties and has been investigated as a potential therapy for various autoimmune diseases. [12] Previous studies have shown that natural *Schistosoma* infection or exposure to SEA can alleviate colitis by modulating the host immune response. [6,28,29,30] However, the specific active components within the SEA responsible for this effect remained unknown, considering the fact that SEA is a mixture of many different proteins. [16] In this study, we identified that both the crude SEA extract and recombinant Smp40 alleviated DSS-induced colitis, whereas recombinant SM14 exacerbated the disease. At the same time, recombinant GST28 had no significant effect on colitis (Figs 2-4). Further investigation revealed that the observed differences in the IBD outcomes may be due to the modulation of immune response by different antigens, albeit in distinct ways (Figs 5-7).

Both naturally-occurring IBD and experimentally-induced colitis are driven by an altered and imbalanced immune response, typically characterised by an

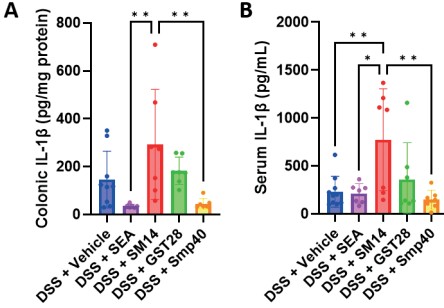

Fig. 4: SM14 exacerbates colonic inflammation in dextran sulphate sodium (DSS)-induced mice. (A) Colonic IL-1β levels. (B) Serum IL-1β levels. *n* = 7-11 mice. Data are presented as mean ± standard deviation (SD). *p < 0.05 and **p < 0.01. Significance determined by one-way analysis of variance (ANOVA).

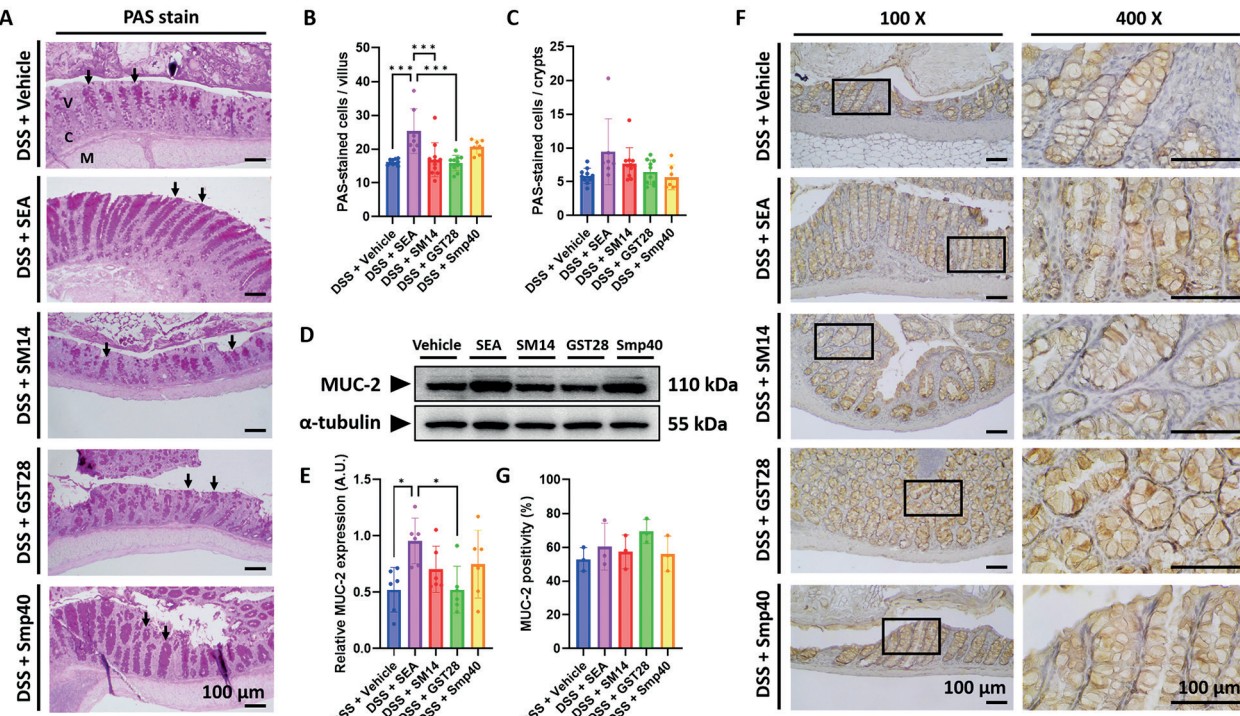

Fig. 5: schistosome egg antigen alters intestinal barrier in mice with dextran sulphate sodium (DSS)-induced colitis. (A) Representative periodic acid-Schiff (PAS)-stained colon section. V: villus; C: crypts; M: muscularis mucosae. The black arrows indicate goblet cells containing mucin (PAS-stained cells), which appear as darker pink-coloured cells. (B-C) Quantification of PAS-stained cells per (B) villus and (C) crypt. (D) Representative western blot image of MUC-2 expression. (E) Relative protein expression of MUC-2. Densitometric values were normalised to α-tubulin. (F) Representative immunohistochemistry staining of MUC-2. (G) Percentage of positive expression of MUC-2. For (A-C), *n* = 7-11 mice; for (D-E), *n* = 6 mice; for (F-G), *n* = 3 mice. Data are presented as mean ± standard deviation (SD). *p < 0.05 and ***p < 0.001. Significance determined by one-way analysis of variance (ANOVA).

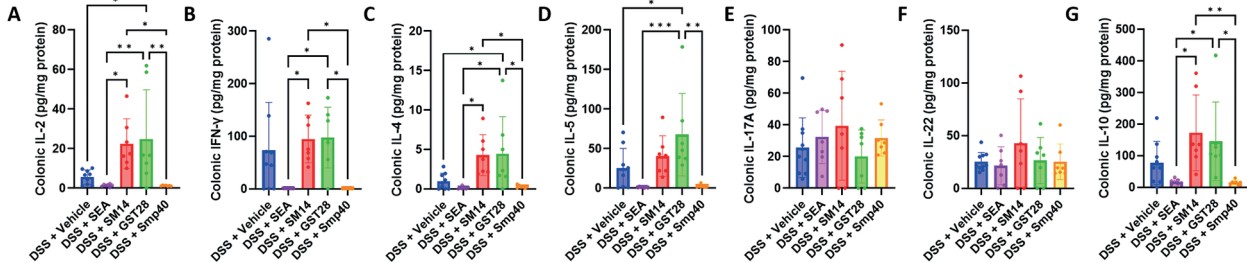

Fig. 6: schistosome egg antigen differently modulates colonic immune responses in dextran sulphate sodium (DSS)-induced mice. Colonic levels of (A) IL-2, (B) IFN-γ, (C) IL-4, (D) IL-5, (E) IL-17A, (F) IL-22, and (G) IL-10. Cytokine levels were relative to the total protein levels. *n* = 7-10 mice. Data are presented as mean ± standard deviation (SD). *p < 0.05; **p < 0.01; and ***p < 0.001. Significance determined by one-way analysis of variance (ANOVA).

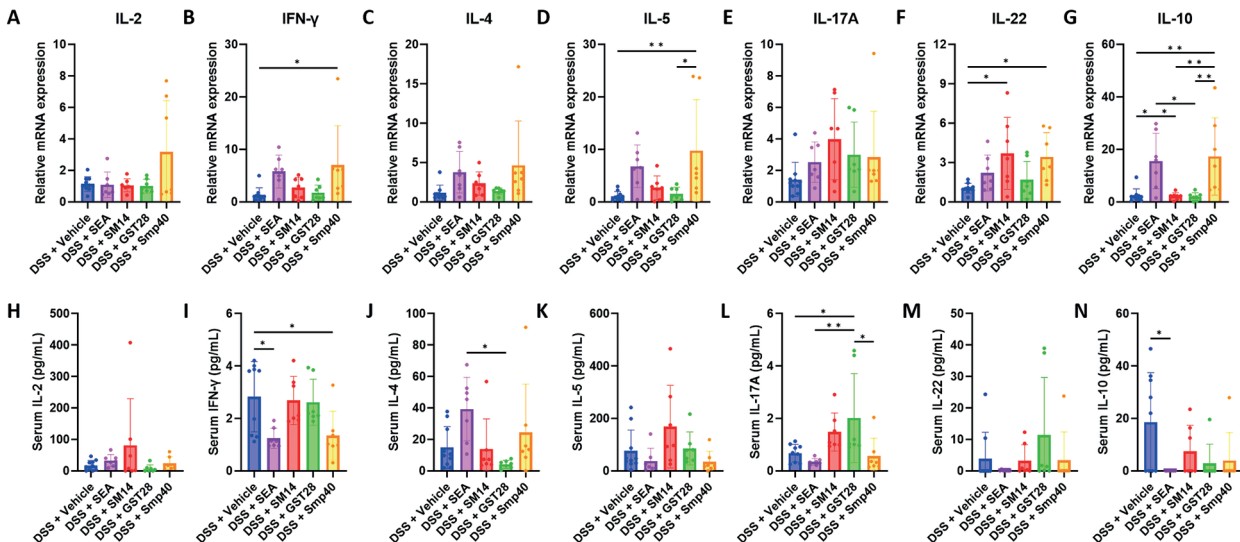

Fig. 7: schistosome egg antigen differently modulates splenic and serum immune responses in dextran sulphate sodium (DSS)-induced mice. Splenic mRNA expression of (A) IL-2, (B) IFN-γ, (C) IL-4, (D) IL-5, (E) IL-17A, (F) IL-22, and (G) IL-10. Serum levels of (H) IL-2, (I) IFN-γ, (J) IL-4, (K) IL-5, (L) IL-17A, (M) IL-22, and (N) IL-10. *n* = 7-10 mice. Data are presented as mean ± standard deviation (SD). *p < 0.05 and **p < 0.01. Significance determined by one-way analysis of variance (ANOVA).

exaggerated Th1 response.[31] Therefore, suppressing this skewed immunity has been considered a potential therapeutic strategy. Treatment of colitic mice with SEA and Smp40 resulted in a significant reduction of colonic Th1 cytokines, IL-2 and IFN-γ (Fig. 6A-B), which was accompanied by an improvement of the colitis (Figs 2-3). This result is consistent with previous studies showing that SEA can suppress Th1 cytokines, leading to the amelioration of autoimmune diseases.[32] However, Th2 cytokines, IL-4 and IL-5, and Treg cytokine IL-10 were also suppressed in the colons of mice treated with SEA and Smp40 (Fig. 6C-D, G). Although SEA is generally known to induce a Th2 response,[33] it has also been suggested that SEA might lead to a substantial reduction of Th2 response under certain inflammatory settings, including colitis.[30] This effect could possibly be due to a pre-existing Th1-dominant immune milieu that constrains SEA from driving an excessive Th2 immune response, thereby restoring Th1 and Th2 balance rather than amplifying Th2 immune responses. The effect of IL-10, a multi-functional cytokine, in IBD is

controversial. Although IL-10 has been shown to have a protective effect in IBD,[33] elevated levels were also found in IBD patients and could contribute to disease risk.[34,35] Previous studies have suggested that *S. mansoni* infection can suppress IL-10 expression, leading to protection from colitis.[5] Similarly, the suppression of IL-10 by SEA and Smp40 observed in this study may contribute to a similar protection. In addition, SEA has been suggested to modulate the gut microbiome and intestinal metabolism in colitic mice, suggesting that the gut microbiome may play a role in SEA-mediated inhibition of inflammation in IBD.[36]

On the other hand, Smp40 is an immunomodulatory protein homologous to heat shock protein (HSP),[21] and several HSPs have already been identified in *Schistosoma* species.[37] Although there is no direct evidence that schistosome-related HSPs influence IBD, various studies have shown that HSPs, such as HSP70, may help ameliorate the disease.[38] Future studies focusing on the protein and molecular characterisation of the Smp40 may provide further knowledge into its role in IBD.

In colitic mice treated with SM14, increased colonic levels of IL-2, IL-4, and IL-10 were observed (Fig. 6A, C, G). Notably, SM14 led to a more pronounced inflammation and the development of colonic dysplasia (Fig. 3A), a known risk factor for colorectal cancer.[39] These findings therefore underscore the potential adverse effects of SM14. We hypothesise that SM14 promotes significant inflammatory cell infiltration and elevated colonic IL-2 levels, which together create a pro-inflammatory environment that may contribute to the development of dysplasia. The increase of IL-4, on the other hand, is crucial for the development of colorectal cancer.[40,41] It has been shown to induce epithelial-mesenchymal transition and promote the aggressiveness of colorectal cancer cells.[42,43] Although IL-10 is generally known for its potential to suppress inflammation and control tumour-promoting inflammation,[44] it has also been associated with colorectal cancer development.[45] In fact, deficiency of IL-10 has been reported to enhance the efficacy of dendritic cell-based immunotherapy,[45] and increased serum IL-10 levels in colorectal cancer patients have been associated with a higher recurrence rate and poorer prognosis.[34] Furthermore, SM14 is a fatty acid-binding protein.[12] While intestinal fatty acid-binding protein that primarily expressed on intestinal epithelial cells plays a pivotal role in intestinal inflammation,[46] exogenous injection of SM14, although differing in their host sources, may also elicit a similar adverse effect. Collectively, these results suggest that SM14 may exacerbate colitis and potentially promote progression toward colorectal cancer. Although SM14 has been used as a vaccine candidate against schistosomiasis and has progressed through phase I and phase II clinical trials with a safe and strong immunogenic profile,[22,47] caution should be exercised in the future when considering its use in IBD patients.

We also observed that GST28-treated mice had increased colonic levels of IL-2, IL-4, and IL-5 (Fig. 6A, C-D); however, no changes in colitis severity were seen (Fig. 2). The therapeutic potential of GST28 in IBD has already been investigated in several experimental and clinical studies.[24,25,26] It has been suggested that GST28 enhances colonic Th2 response, recruits eosinophils, and suppresses Th1 response in order to alleviate colitic symptoms.[24,25] Our findings of increased IL-4 and IL-5 are consistence with these previously reported effects.[24] Previously, GST28 has been shown to increase IL-2 expression in *S. mansoni* infection.[27] Given that IL-2 is a cytokine that is important for the expansion and function of Treg cells,[48] its elevation may provide immunoregulatory benefits. In addition, low-dose IL-2 treatment has been shown to reduce disease severity in 2,4-dinitrobenzene sulfonic acid (DNBS)-induced colitic mice[49] and in patients with moderate to severe ulcerative colitis.[50] Despite the changes in the immune response, GST28 treatment did not improve IBD outcomes in our study. One possible reason is the absence of an adjuvant, which may be necessary to fully activate the anti-inflammatory effect of GST28.[24,51] Additionally, dosing of GST28 appears to be an important factor. In a previous study, the use of GST28 at doses of 5 and 50 μg/kg significantly suppressed colitis, whereas a higher dose of 500 μg/kg

had no effect on colitis.[25] In our current study, the dosage used (250 μg/mL, approximately equal to 830 μg/kg) may have exceeded the effective therapeutic range. However, our previous work showed that GST28 at this same dose, when combined with heat-killed *Cutibacterium acnes* as an adjuvant, induced significant immune modulation and reduced disease severity of schistosomiasis,[27] emphasising the importance of adjuvant inclusion. Therefore, future studies may involve the combination use of GST28 with appropriate adjuvants in the context of IBD. The use of adjuvants may also be applied and investigated for other egg antigens to enhance their immune modulatory potential.

Th17 responses are also involved in the pathogenesis of IBD by promoting IL-17-mediated intestinal inflammation.[52] However, our study did not reveal any significant changes in the intestinal Th17 response following treatment with any of the tested egg antigens (Fig. 6E-F). Previous studies have suggested that SEA can protect against skin transplant rejection by modulating Th1 and Th2 responses, without affecting IL-17+CD4+ T cells.[53] Similarly, SM14 has been shown to drive the development of IL-10-producing T cells, but not IL-17-producing T cells, in a C57BL/6 mouse model.[18] Moreover, the choice of mouse strain may also influence the immune outcome. A prior study comparing *Giardia* infection in BALB/c and C57BL/6 mice showed a higher parasitic burden in BALB/c mice, which is associated with a lower Th17 activity.[54] Another report suggested that *Mycoplasma pneumoniae* infection in BALB/c mice exhibited a lower Th17 response compared to DBA/2 mice.[55] Therefore, while Th17 responses play a role in IBD, the tested egg antigens may not appear to significantly influence colonic Th17 activity. Additionally, variations in immune responses among different mouse strains highlight the importance of host genetic background in interpreting antigen-specific immunological outcomes in IBD.[56,57] Therefore, future studies comparing different mouse models may be warranted.

Finally, we investigated the systemic immune response in the mice, which the results differed from the local response as observed in the colon. Smp40-treated mice showed a significant increase in splenic IL-2, IFN-γ, IL-5, IL-22, and IL-10 levels, whereas SEA-treated mice had higher splenic IL-10 levels, and SM14-treated mice had higher splenic IL-22 levels (Fig. 7A-G). A previous study has shown that Smp40 can elicit a Th1 response, characterised by increased secretion of IL-2 and IFN-γ from splenic lymphocytes, even as the overall immune response shifts toward Th2 dominance.[58] Additionally, Smp40 has been shown to increase IL-10 expression in peripheral blood mononuclear cells from *S. mansoni*-infected patients.[21] SM14 induction of splenic IL-22 may contribute to the excess inflammation seen in the colon (Fig. 3A), as IL-22 is known to recruit neutrophils and promote an inflammatory environment in ulcerative colitis.[59] SEA has been shown to stimulate IL-10 production in splenic B cells, which aligns with our results.[18] From the same study, it was suggested that glycosylated molecules in high molecular weight fractions of SEA are responsible for this effect,[18] suggesting a specific component may underlie its immunomodulatory effect.

In this study, different egg antigens demonstrated different immunomodulatory effects in the context of IBD, leading to varying disease outcomes. While many studies have highlighted the promising therapeutic potential of SEA in treating IBD, it is important to recognise that SEA is a complex mixture of many different molecules, each of which may exert both beneficial and detrimental effects. Therefore, a deeper understanding of the safety and mechanisms of these individual antigens is necessary before considering their use in clinical settings.

## AUTHORS' CONTRIBUTION

HWF - investigation, formal analysis, writing-review & editing; HYPL - conceptualisation, investigation, funding acquisition, writing-review & editing. The authors have declared that no competing interests exist.

## DATA AVAILABILITY

All relevant data have been included in the manuscript and its supporting information files, or have been deposited in Mendeley Data (doi: 10.17632/rcpnbrx2hv.1).

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

# OPEN PEER REVIEW

Memórias do IOC thanks the anonymous reviewers for their contribution to the peer review of this work.

## FIRST REVIEW ROUND

REVIEWERS' COMMENTS

### REVIEWER #1

The work aims to investigate the role of S. mansoni egg antigens as a potential therapeutic agent for inflammatory bowel disease.

The work has relevance and interesting and original results.

Adequacy of the abstract:  The methodology wasn't fully explored in the abstract and doesn't include the methodological steps described in the article. Even if it's brief, I suggest rewriting it to include the techniques used.

The methodology failed to describe how the antigen injection was administered during the week. Was it administered three times a week on alternate days? Three days in a row? Explain this step further.

The results are well written but I suggest changing the figures. Separate the images from the graphs in Figure 2. The histopathology images are very small. This prevents visualization of the structures. For example, it is impossible to see the inflammatory process with such small images. I suggest creating a Figure containing the images of the histopathological sections and the macroscopic view of the intestine (C) occupying one page.Separate the images from the graphs in Figure 2. The histopathology images are very small. This prevents visualization of the structures. For example, it is impossible to see the inflammatory process with such small images. I suggest creating a Figure containing the images of the histopathological sections and the macroscopic view of the intestine (C) occupying one page.

In figure 3A you need to identify the structures observed and described in the results.

The discussion is well-founded and discusses the main results obtained.

References are relevant and current.

### REVIEWER #2

a) Adequacy of the abstract:

The abstract is clear and concise, presenting the objectives, methodology, main results, and conclusions of the study well. It adequately summarizes the distinct effect of different soluble egg antigen (SEA) proteins in a murine model of colitis. However, it could be enriched with epidemiological data and public health impacts brought about by both inflammatory bowel disease (IBD) and schistosomiasis.

b) Originality and importance of the contribution development of the field of study:

The study is original in that it separately evaluates SEA components in the modulation of inflammatory bowel disease (IBD), a relevant and under-explored topic. Its findings are important for the advancement of natural immunomodulation and helminth-based therapies, offering valuable insights for future translational research.

c) Relevance of methodology, results, and discussion:

The methodology is adequate, using a valid murine model for experimental colitis, with detailed immunological, histological, and molecular analyses.

The article does not explicitly detail the sample calculation for the number of animals used in the different groups evaluated with soluble Schistosoma egg antigens (SEA). I believe that without this information, it is difficult to assess the statistical robustness of the study.

The results are consistent, highlighting important differences between the antigens. The discussion engages well with the current literature, recognizing limitations and challenges for clinical translation.

d) References:

The references are current and relevant to the topic - schistosomiasis and inflammatory diseases

e) Figures and tables:

The figures and tables are well designed, clear and adequately illustrate the data. I suggest improving Figure 1, especially the proportion of arrows and syringes indicating peritoneal injection of antigens so that they fit within the bounding spaces that symbolize the weeks of the study.

### EDITOR COMMENTS:

1. Line 28: "250 µg/mL crude SEA extract and recombinant egg antigen proteins" change to " 250 µg/mL crude SEA extract or recombinant egg antigen protein".

2. Line 218: "a week (Figure 1A)." and Line 325: "colonic dysplasia (Figure 1E)," Figure 1 has no letter divisions.

3. Fig 5 A, as the p value is not significant, this can be described in the text in the section "Schistosome egg antigen differently modulates immune responses of colitic mice", with the exact value found, but it should be removed from the image.

4. Line 118: "The plasmid of the Smp40 was newly designed and constructed in this experiment." and Lines 137-138: "The eluted proteins were analyzed on a 13.5% SDS-PAGE and stained with 0.1% Coomassie blue."

A supplementary figure containing design of plasmid and gel with and without IPTG lysate, as well as remaining pellet is necessary. Also, did the authors do a western blot tagging the histidine tail?

## AUTHORS' RESPONSE TO THE REVIEWERS

Dear Editor and Reviewers,

Thank you very much for having considered our manuscript. We are very happy to have received a positive evaluation, and we would like to express our appreciation for your thoughtful comments and helpful suggestions. We fundamentally agree with all the comments made by the reviewers, and we have incorporated corresponding revisions into the manuscript. Our detailed, point-by-point responses to the editorial and reviewer comments are given below.

Reviewer: 1

Reviewer comments: The work aims to investigate the role of S. mansoni egg antigens as a potential therapeutic agent for inflammatory bowel disease.

The work has relevance and interesting and original results.

Adequacy of the abstract: The methodology wasn't fully explored in the abstract and doesn't include the methodological steps described in the article. Even if it's brief, I suggest rewriting it to include the techniques used.

Response: Thank you for the comments. We have added information about the techniques used to the abstract section.

Line 32-34: *"Colonic histopathology was assessed by H&E staining, and the immune response was evaluated through periodic acid-Schiff (PAS) staining, immunohistochemistry, ELISA, western blot, and qPCR."*

The methodology failed to describe how the antigen injection was administered during the week. Was it administered three times a week on alternate days? Three days in a row? Explain this step further.

Response: Thank you for pointing this out. The mice were injected with antigens three times a week on alternate days (Monday, Wednesday, and Friday). We have added this information to the methodology.

Line 161-163: *"Different egg antigen proteins (at a concentration of 250 µg/mL) were peritoneally injected into the mice three times a week on alternate days during the experimental duration."*

The results are well written but I suggest changing the figures. Separate the images from the graphs in Figure 2. The histopathology images are very small. This prevents visualization of the structures. For example, it is impossible to see the inflammatory process with such small images. I suggest creating a Figure containing the images of the histopathological sections and the macroscopic view of the intestine (C) occupying one page.

Response: Thank you for pointing this out. We agree with the reviewer that the histopathology images were too small in the previous version. To improve readability, we have modified the figures. The original Figure 2 has been separated into three individual figures, with the new Figure 2 showing the body weight and the gross image of the intestine, Figure 3 showing the histopathology, and Figure 4 showing the results of IL-1β.

In addition, in the PDF version of the review file, the figures were compressed to fit the page, which may have also contributed to the reduced image clarity.

In figure 3A you need to identify the structures observed and described in the results.

The discussion is well-founded and discusses the main results obtained.

References are relevant and current.

Response: Thank you for the comments. We have added the labelling in the figure (the new Figure 5A) and have included a description of each label in the figure legend. We have also added a corresponding description in the result section (lines 262-267).

Figure Legend 5A: *"(A) Representative Periodic acid-Schiff (PAS)-stained colon section. V, villus; C, crypts; M, muscularis mucosae. The black arrows indicate goblet cells containing mucin (PAS-stained cells), which appear as darker pink-colored cells."*

Line 265-270: *"To further investigate the function of the intestinal barrier, PAS staining was used to identify mucin within goblet cells, which appeared dark pink in contrast to the pale pink background. The result suggested that SEA-treated mice had significantly higher colonic goblet cells in the villus compared to the vehicle and other treatment groups (Figure 5A and B). Yet, goblet cell numbers were not significantly different in the crypt (Figure 5C)."*

Reviewer: 2

Reviewer comments:

a) Adequacy of the abstract:

The abstract is clear and concise, presenting the objectives, methodology, main results, and conclusions of the study well. It adequately summarizes the distinct effect of different soluble egg antigen (SEA) proteins in a murine model of colitis. However, it could be enriched with epidemiological data and public health impacts brought about by both inflammatory bowel disease (IBD) and schistosomiasis.

Response: Thank you for the comments. Due to the word-count limitation, we have only added some brief information in the abstract. However, additional details have been added in the introduction section (lines 55-58 and 70-72).

Abstract, BACKGROUND: *"Inflammatory bowel disease (IBD) is an increasingly prevalent disease, affecting over seven million people worldwide and imposes a heavy burden on public health. The rising prevalence of IBD may be attributed to the hygiene hypothesis, which suggests that reduced exposure to parasites and microbes may weaken the immune system, thereby increasing susceptibility to developing IBD... ."*

Line 55-58: *"Inflammatory bowel disease (IBD) is an increasingly prevalent chronic inflammatory condition that currently affects over seven million people worldwide. The disease places a substantial burden on public health, leading to high medical costs, reduced productivity, and diminished quality of life."*

Line 70-72: *"Among the diseases caused by these parasites, schistosomiasis ranks as the second most important parasitic disease after malaria, affecting more than 200 million people around the world."*

b) Originality and importance of the contribution development of the field of study:

The study is original in that it separately evaluates SEA components in the modulation of inflammatory bowel disease (IBD), a relevant and under-explored topic. Its findings are important for the advancement of natural immunomodulation and helminth-based therapies, offering valuable insights for future translational research.

Response: We sincerely thank the reviewer for the positive assessment of our work.

c) Relevance of methodology, results, and discussion:

The methodology is adequate, using a valid murine model for experimental colitis, with detailed immunological, histological, and molecular analyses.

The article does not explicitly detail the sample calculation for the number of animals used in the different groups evaluated with soluble Schistosoma egg antigens (SEA). I believe that without this information, it is difficult to assess the statistical robustness of the study.

Response: Thank you for highlighting this point. The sample size was calculated by the resource equation, resulting in the allocation of three to five mice per group. The experiment was independently repeated twice to validate the results. For data analysis, results from both independent experiments were pooled, and a statistical comparison was performed on the combined dataset without applying interim statistical adjustments. We have added this information in the methodology section (lines 155-158 and 223-225).

Line 155-158: *"Mice were divided randomly into five groups. The sample size was calculated using the resource equation, resulting in the allocation of three to five mice per group. The experiment was independently repeated twice."*

Line 223-225: *"Two independent experimental replicates were performed. Data from the same groups across both repeats were pooled, and a statistical comparison was performed on the combined dataset without applying interim statistical adjustments."*

The results are consistent, highlighting important differences between the antigens. The discussion engages well with the current literature, recognizing limitations and challenges for clinical translation.

Response: We sincerely thank the reviewer for the positive assessment of our work.

d) References:

The references are current and relevant to the topic - schistosomiasis and inflammatory diseases

Response: We sincerely thank the reviewer for the positive assessment of our work.

e) Figures and tables:

The figures and tables are well designed, clear and adequately illustrate the data. I suggest improving Figure 1, especially the proportion of arrows and syringes indicating peritoneal injection of antigens so that they fit within the bounding spaces that symbolize the weeks of the study.

Response: Thank you for the comments. We have revised Figure 1 to improve clarity and readability.

**EDITOR COMMENTS:**

1. Line 28: "250 μg/mL crude SEA extract and recombinant egg antigen proteins" change to " 250 μg/mL crude SEA extract or recombinant egg antigen protein".

Response: Thank you for the comments. It has been revised.

Line 30-32: *"Throughout the experiment, mice were intraperitoneally injected with 250 μg/mL crude SEA extract or recombinant egg antigen proteins, including SM14, GST28, and SMP40, three times a week."*

2. Line 218: "a week (Figure 1A)." and Line 325: "colonic dysplasia (Figure 1E)," Figure 1 has no letter divisions.

Response: Thank you for pointing out these mistakes. These have been corrected.

3. Fig 5 A, as the p value is not significant, this can be described in the text in the section "Schistosome egg antigen differently modulates immune responses of colitic mice", with the exact value found, but it should be removed from the image.

Response: Thank you for the comments. The p-value has been removed from the figure. The exact value was also added in the content (lines 291-293).

Line 291-293: *"IL-2, although not showing statistical significance (p = 0.0597), also revealed a similar increase in the Smp40-treated group (Figure 7A)."*

4. Line 118: "The plasmid of the Smp40 was newly designed and constructed in this experiment." and Lines 137-138: "The eluted proteins were analyzed on a 13.5% SDS-PAGE and stained with 0.1% Coomassie blue."

A supplementary figure containing design of plasmid and gel with and without IPTG lysate, as well as remaining pellet is necessary. Also, did the authors do a western blot tagging the histidine tail?

Response: Thank you for your comments. We have added a supplementary figure that includes: (Supp Fig 1A) the plasmid design, (Supp Fig 1B) an SDS-PAGE gel of the lysate and eluted proteins, and (Supp Fig 1C) a western blot showing detection of the histidine tag. The information has been added in the methodology part (lines 137-138 and lines 151-153) and in the Figure legend.

Line 137-138: *"The resulting product was digested with NdeI and XhoI and ligated into plasmid pET-30a (Supplementary Figure 1A)."*

Line 151-153: *"An SDS-PAGE gel showing the bacterial lysates for Smp40 with and without IPTG induction, as well as the eluted Smp40 protein, is presented in Supplementary Figure 1B-D."*

Figure Legend:

Supplementary Figure 1. *Characterization of the newly designed Smp40 protein used in this study. (A) Schematic representation of the plasmid construct encoding Smp40. (B) SDS-PAGE analysis of bacterial lysates with or without IPTG induction. L, protein ladder; lane 1, lysate without IPTG induction; lane 2, lysate after 4 h of IPTG induction. (C) SDS-PAGE analysis of protein purification steps. L, protein ladder; lane 1, supernatant fraction; lane 2, debris fraction; lane 3, flow-through fraction; lane 4, wash fraction; lane 5, eluted protein. (D) Western blot analysis detecting the histidine tag on the purified Smp40 protein.*

## SECOND REVIEW ROUND

### REVIEWERS' COMMENTS

### REVIEWER #1

After review, all my suggestions were added and changed in the manuscript. Now, all items are suitable for publication.

### REVIEWER #2

I have reviewed the authors' responses to my previous comments and the corresponding revisions made to the manuscript. I am satisfied with the comprehensive and thoughtful way in which the authors have addressed the points raised.

The additional information provided regarding the epidemiological context of IBD and schistosomiasis, the clarification of sample size calculation and replication, as well as the improvements in Figure 1, have considerably strengthened the manuscript.

The revisions have enhanced the clarity, methodological transparency, and overall scientific quality of the study.

