## [Reviewer Report · FIRST REVIEW ROUND - REVIEWERS' COMMENTS]

## REVIEWER #1

The work aims to investigate the role of *S. mansoni* egg antigens as a potential therapeutic agent for inflammatory bowel disease.

The work has relevance and interesting and original results.

**Adequacy of the abstract:** The methodology wasn't fully explored in the abstract and doesn't include the methodological steps described in the article. Even if it's brief, I suggest rewriting it to include the techniques used.

The methodology failed to describe how the antigen injection was administered during the week. Was it administered three times a week on alternate days? Three days in a row? Explain this step further.

The results are well written but I suggest changing the figures. Separate the images from the graphs in Figure 2. The histopathology images are very small. This prevents visualization of the structures. For example, it is impossible to see the inflammatory process with such small images. I suggest creating a Figure containing the images of the histopathological sections and the macroscopic view of the intestine (C) occupying one page. Separate the images from the graphs in Figure 2. The histopathology images are very small. This prevents visualization of the structures. For example, it is impossible to see the inflammatory process with such small images. I suggest creating a Figure containing the images of the histopathological sections and the macroscopic view of the intestine (C) occupying one page.

In figure 3A you need to identify the structures observed and described in the results.

The discussion is well-founded and discusses the main results obtained.

References are relevant and current.

## REVIEWER #2

**a) Adequacy of the abstract:**

The abstract is clear and concise, presenting the objectives, methodology, main results, and conclusions of the study well. It adequately summarizes the distinct effect of different soluble egg antigen (SEA) proteins in a murine model of colitis. However, it could be enriched with epidemiological data and public health impacts brought about by both inflammatory bowel disease (IBD) and schistosomiasis.

**b) Originality and importance of the contribution development of the field of study:**

The study is original in that it separately evaluates SEA components in the modulation of inflammatory bowel disease (IBD), a relevant and under-explored topic. Its findings are important for the advancement of natural immunomodulation and helminth-based therapies, offering valuable insights for future translational research.

**c) Relevance of methodology, results, and discussion:**

The methodology is adequate, using a valid murine model for experimental colitis, with detailed immunological, histological, and molecular analyses.

The article does not explicitly detail the sample calculation for the number of animals used in the different groups evaluated with soluble *Schistosoma* egg antigens (SEA). I believe that without this information, it is difficult to assess the statistical robustness of the study.

The results are consistent, highlighting important differences between the antigens. The discussion engages well with the current literature, recognizing limitations and challenges for clinical translation.

**d) References:**

The references are current and relevant to the topic - schistosomiasis and inflammatory diseases

**e) Figures and tables:**

The figures and tables are well designed, clear and adequately illustrate the data. I suggest improving Figure 1, especially the proportion of arrows and syringes indicating peritoneal injection of antigens so that they fit within the bounding spaces that symbolize the weeks of the study.

## EDITOR COMMENTS:

1. Line 28: "250 μg/mL crude SEA extract and recombinant egg antigen proteins" change to " 250 μg/mL crude SEA extract or recombinant egg antigen protein".

2. Line 218: "a week (Figure 1A)." and Line 325: "colonic dysplasia (Figure 1E)," Figure 1 has no letter divisions.

3. Fig 5 A, as the p value is not significant, this can be described in the text in the section "Schistosome egg antigen differently modulates immune responses of colitic mice", with the exact value found, but it should be removed from the image.

4. Line 118: "The plasmid of the Smp40 was newly designed and constructed in this experiment." and Lines 137-138: "The eluted proteins were analyzed on a 13.5% SDS-PAGE and stained with 0.1% Coomassie blue."

A [Supplementary-material s1] containing design of plasmid and gel with and without IPTG lysate, as well as remaining pellet is necessary. Also, did the authors do a western blot tagging the histidine tail?

## AUTHORS' RESPONSE TO THE REVIEWERS

Dear Editor and Reviewers,

Thank you very much for having considered our manuscript. We are very happy to have received a positive evaluation, and we would like to express our appreciation for your thoughtful comments and helpful suggestions. We fundamentally agree with all the comments made by the reviewers, and we have incorporated corresponding revisions into the manuscript. Our detailed, point-by-point responses to the editorial and reviewer comments are given below.

**Reviewer: 1**

**Reviewer comments:** The work aims to investigate the role of *S. mansoni* egg antigens as a potential therapeutic agent for inflammatory bowel disease.

The work has relevance and interesting and original results.

Adequacy of the abstract: The methodology wasn't fully explored in the abstract and doesn't include the methodological steps described in the article. Even if it's brief, I suggest rewriting it to include the techniques used.

**Response:** Thank you for the comments. We have added information about the techniques used to the abstract section.

Line 32-34: "Colonic histopathology was assessed by H&E staining, and the immune response was evaluated through periodic acid-Schiff (PAS) staining, immunohistochemistry, ELISA, western blot, and qPCR."

**Reviewer comments:** The methodology failed to describe how the antigen injection was administered during the week. Was it administered three times a week on alternate days? Three days in a row? Explain this step further.

**Response:** Thank you for pointing this out. The mice were injected with antigens three times a week on alternate days (Monday, Wednesday, and Friday). We have added this information to the methodology.

Line 161-163: "Different egg antigen proteins (at a concentration of 250 μg/mL) were peritoneally injected into the mice three times a week on alternate days during the experimental duration."

**Reviewer comments:** The results are well written but I suggest changing the figures. Separate the images from the graphs in Figure 2. The histopathology images are very small. This prevents visualization of the structures. For example, it is impossible to see the inflammatory process with such small images. I suggest creating a Figure containing the images of the histopathological sections and the macroscopic view of the intestine (C) occupying one page.

**Response:** Thank you for pointing this out. We agree with the reviewer that the histopathology images were too small in the previous version. To improve readability, we have modified the figures. The original Figure 2 has been separated into three individual figures, with the new Figure 2 showing the body weight and the gross image of the intestine, Figure 3 showing the histopathology, and Figure 4 showing the results of IL-1β.

In addition, in the PDF version of the review file, the figures were compressed to fit the page, which may have also contributed to the reduced image clarity.

**Reviewer comments:** In figure 3A you need to identify the structures observed and described in the results.

The discussion is well-founded and discusses the main results obtained.

References are relevant and current.

**Response:** Thank you for the comments. We have added the labelling in the figure (the new Figure 5A) and have included a description of each label in the figure legend. We have also added a corresponding description in the result section (lines 262-267).

Figure Legend 5A: "(A) Representative Periodic acid-Schiff (PAS)-stained colon section. V, villus; C, crypts; M, muscularis mucosae. The black arrows indicate goblet cells containing mucin (PAS-stained cells), which appear as darker pink-colored cells."

Line 265-270: "To further investigate the function of the intestinal barrier, PAS staining was used to identify mucin within goblet cells, which appeared dark pink in contrast to the pale pink background. The result suggested that SEA-treated mice had significantly higher colonic goblet cells in the villus compared to the vehicle and other treatment groups (Figure 5A and B). Yet, goblet cell numbers were not significantly different in the crypt (Figure 5C)."

**Reviewer: 2**

**Reviewer comments:**

**a) Adequacy of the abstract:**

The abstract is clear and concise, presenting the objectives, methodology, main results, and conclusions of the study well. It adequately summarizes the distinct effect of different soluble egg antigen (SEA) proteins in a murine model of colitis. However, it could be enriched with epidemiological data and public health impacts brought about by both inflammatory bowel disease (IBD) and schistosomiasis.

**Response:** Thank you for the comments. Due to the word-count limitation, we have only added some brief information in the abstract. However, additional details have been added in the introduction section (lines 55-58 and 70-72).

Abstract, BACKGROUND: "Inflammatory bowel disease (IBD) is an increasingly prevalent disease, affecting over seven million people worldwide and imposes a heavy burden on public health. The rising prevalence of IBD may be attributed to the hygiene hypothesis, which suggests that reduced exposure to parasites and microbes may weaken the immune system, thereby increasing susceptibility to developing IBD… . "

Line 55-58: "Inflammatory bowel disease (IBD) is an increasingly prevalent chronic inflammatory condition that currently affects over seven million people worldwide. The disease places a substantial burden on public health, leading to high medical costs, reduced productivity, and diminished quality of life."

Line 70-72: "Among the diseases caused by these parasites, schistosomiasis ranks as the second most important parasitic disease after malaria, affecting more than 200 million people around the world."

**b) Originality and importance of the contribution development of the field of study:**

The study is original in that it separately evaluates SEA components in the modulation of inflammatory bowel disease (IBD), a relevant and under-explored topic. Its findings are important for the advancement of natural immunomodulation and helminth-based therapies, offering valuable insights for future translational research.

**Response:** We sincerely thank the reviewer for the positive assessment of our work.

**c) Relevance of methodology, results, and discussion:**

The methodology is adequate, using a valid murine model for experimental colitis, with detailed immunological, histological, and molecular analyses.

The article does not explicitly detail the sample calculation for the number of animals used in the different groups evaluated with soluble *Schistosoma* egg antigens (SEA). I believe that without this information, it is difficult to assess the statistical robustness of the study.

**Response:** Thank you for highlighting this point. The sample size was calculated by the resource equation, resulting in the allocation of three to five mice per group. The experiment was independently repeated twice to validate the results. For data analysis, results from both independent experiments were pooled, and a statistical comparison was performed on the combined dataset without applying interim statistical adjustments. We have added this information in the methodology section (lines 155-158 and 223-225).

Line 155-158: "Mice were divided randomly into five groups. The sample size was calculated using the resource equation, resulting in the allocation of three to five mice per group. The experiment was independently repeated twice."

Line 223-225: "Two independent experimental replicates were performed. Data from the same groups across both repeats were pooled, and a statistical comparison was performed on the combined dataset without applying interim statistical adjustments."

**Reviewer comments:** The results are consistent, highlighting important differences between the antigens. The discussion engages well with the current literature, recognizing limitations and challenges for clinical translation.

**Response:** We sincerely thank the reviewer for the positive assessment of our work.

**d) References:**

The references are current and relevant to the topic - schistosomiasis and inflammatory diseases

**Response:** We sincerely thank the reviewer for the positive assessment of our work.

**e) Figures and tables:**

The figures and tables are well designed, clear and adequately illustrate the data. I suggest improving Figure 1, especially the proportion of arrows and syringes indicating peritoneal injection of antigens so that they fit within the bounding spaces that symbolize the weeks of the study.

**Response:** Thank you for the comments. We have revised Figure 1 to improve clarity and readability.

**EDITOR COMMENTS:**

**1.** Line 28: "250 μg/mL crude SEA extract and recombinant egg antigen proteins" change to " 250 μg/mL crude SEA extract or recombinant egg antigen protein".

**Response:** Thank you for the comments. It has been revised.

Line 30-32: "Throughout the experiment, mice were intraperitoneally injected with 250 μg/mL crude SEA extract or recombinant egg antigen proteins, including SM14, GST28, and SMP40, three times a week."

**2.** Line 218: "a week (Figure 1A)." and Line 325: "colonic dysplasia (Figure 1E)," Figure 1 has no letter divisions.

**Response:** Thank you for pointing out these mistakes. These have been corrected.

**3.** Fig 5 A, as the p value is not significant, this can be described in the text in the section "Schistosome egg antigen differently modulates immune responses of colitic mice", with the exact value found, but it should be removed from the image.

**Response:** Thank you for the comments. The p-value has been removed from the figure. The exact value was also added in the content (lines 291-293).

Line 291-293: "IL-2, although not showing statistical significance (p = 0.0597), also revealed a similar increase in the Smp40-treated group (Figure 7A)."

**4.** Line 118: "The plasmid of the Smp40 was newly designed and constructed in this experiment." and Lines 137-138: "The eluted proteins were analyzed on a 13.5% SDS-PAGE and stained with 0.1% Coomassie blue."

A [Supplementary-material s1] containing design of plasmid and gel with and without IPTG lysate, as well as remaining pellet is necessary. Also, did the authors do a western blot tagging the histidine tail?

**Response:** Thank you for your comments. We have added a [Supplementary-material s1] that includes: (Supp Fig 1A) the plasmid design, (Supp Fig 1B) an SDS-PAGE gel of the lysate and eluted proteins, and (Supp Fig 1C) a western blot showing detection of the histidine tag. The information has been added in the methodology part (lines 137-138 and lines 151-153) and in the Figure legend.

Line 137-138: "The resulting product was digested with NdeI and XhoI and ligated into plasmid pET-30a ([Supplementary-material s1] 1A)."

Line 151-153: "An SDS-PAGE gel showing the bacterial lysates for Smp40 with and without IPTG induction, as well as the eluted Smp40 protein, is presented in [Supplementary-material s1] 1B-D."

Figure Legend:

[Supplementary-material s1] 1. Characterization of the newly designed Smp40 protein used in this study. (A) Schematic representation of the plasmid construct encoding Smp40. (B) SDS-PAGE analysis of bacterial lysates with or without IPTG induction. L, protein ladder; lane 1, lysate without IPTG induction; lane 2, lysate after 4 h of IPTG induction. (C) SDS-PAGE analysis of protein purification steps. L, protein ladder; lane 1, supernatant fraction; lane 2, debris fraction; lane 3, flow-through fraction; lane 4, wash fraction; lane 5, eluted protein. (D) Western blot analysis detecting the histidine tag on the purified Smp40 protein.

---

## [Reviewer Report · REVIEWERS COMMENTS]

## REVIEWER #1

After review, all my suggestions were added and changed in the manuscript. Now, all items are suitable for publication.

## REVIEWER #2

I have reviewed the authors' responses to my previous comments and the corresponding revisions made to the manuscript. I am satisfied with the comprehensive and thoughtful way in which the authors have addressed the points raised.

The additional information provided regarding the epidemiological context of IBD and schistosomiasis, the clarification of sample size calculation and replication, as well as the improvements in Figure 1, have considerably strengthened the manuscript.

The revisions have enhanced the clarity, methodological transparency, and overall scientific quality of the study.